# Differential Consequences of *Bmp9* Deletion on Sinusoidal Endothelial Cell Differentiation and Liver Fibrosis in 129/Ola and C57BL/6 Mice

**DOI:** 10.3390/cells8091079

**Published:** 2019-09-13

**Authors:** Agnès Desroches-Castan, Emmanuelle Tillet, Nicolas Ricard, Marie Ouarné, Christine Mallet, Jean-Jacques Feige, Sabine Bailly

**Affiliations:** Biology of Cancer and Infection Laboratory, University Grenoble Alpes, Inserm, CEA, F-38000 Grenoble, France; agnes.castan@cea.fr (A.D.-C.); emmanuelle.tillet@cea.fr (E.T.); nic.ricard@gmail.com (N.R.); marie.ouarne@medicina.ulisboa.pt (M.O.); christine.mallet@cea.fr (C.M.); jean-jacques.feige@cea.fr (J.-J.F.)

**Keywords:** BMP9, genetic background, liver, fibrosis, capillarization, liver sinusoidal endothelial cells, HHT, mouse, fenestrae, plvap

## Abstract

The aim of the present work was to address the role of BMP9 in different genetic backgrounds (C57BL/6, BALB/c, and 129/Ola) of mice deleted for *Bmp9*. We found that *Bmp9* deletion led to premature mortality only in the 129/Ola strain. We have previously shown that *Bmp9* deletion led to liver sinusoidal endothelial cells (LSEC) capillarization and liver fibrosis in the 129/Ola background. Here, we showed that this is not the case in the C57BL/6 background. Analysis of LSEC from Wild-type (WT) versus *Bmp9*-KO mice in the C57BL/6 background showed no difference in LSEC fenestration and in the expression of differentiation markers. Comparison of the mRNA expression of LSEC differentiation markers between WT C57BL/6 and 129/Ola mice showed a significant decrease in Stabilin2, Plvap, and CD209b, suggesting a more capillary-like phenotype in WT C57BL/6 LSECs. C57BL/6 mice also had lower BMP9 circulating concentrations and hepatic Vegfr2 mRNA levels, compared to the 129/Ola mice. Taken together, our observations support a role for BMP9 in liver endothelial cell fenestration and prevention of fibrosis that is dependent on genetic background. It also suggests that 129/Ola mice are a more suitable model than C57BL/6 for the study of liver fibrosis subsequent to LSEC capillarization.

## 1. Introduction

BMP9 (Bone morphogenetic protein 9), also known as GDF2 (growth and differentiation factor 2), belongs to the TGFβ family whose members are involved in many different biological activities [1,2]. BMP9 is produced by hepatic stellate cells (HSCs) [3,4]. BMP9 is present in blood at active concentrations and is now recognized to be a potent vascular quiescence factor [4,5]. Recombinant BMP9 was also shown to be one of the most potent osteogenic inducers and a regulator of several biological functions, including chondrogenesis, glucose metabolism, neuronal differentiation, and iron balance [6]. BMP9 is the physiological high affinity ligand (Kd: 2pM) of the endothelial-specific receptor ALK1 (activin receptor-like kinase 1) and of its co-receptor endoglin [7,8,9].

The role of BMP9 in liver homeostasis is still unclear and controversial [10,11]. BMP9 was reported to be an autocrine/paracrine cytokine inhibiting adult (non-malignant) hepatocyte proliferation [3]. Recently, Breitkopf-Heinlein et al. and Addante et al. published that BMP9 is a profibrogenic factor in the liver [3,12]. In line with these works, Li et al. found that higher BMP9 levels accompanied advanced stages of liver fibrosis [13]. In contrast, analysis of distinct transcriptomic datasets from those mentioned above revealed that hepatic BMP9 mRNA expression is decreased in several human liver diseases including cirrhosis, NASH (non-alcoholic steato hepatitis) and hepatocarcinomas [14]. We have recently addressed the role of BMP9 in liver homeostasis by studying the phenotype of *Bmp9* deleted mice in the 129/Ola genetic background [15]. In this previous work, we found that *Bmp9* deletion triggers hepatic perisinusoidal fibrosis by regulating the differentiated state of liver sinusoidal endothelial cells (LSEC), leading to a dedifferentiated state of LSEC. This dedifferentiation state is called capillarization and corresponds to the loss of LSEC-specific markers, loss of LSEC fenestration, and formation of an organized basement membrane in the space of Disse [16]. Capillarization often precedes human liver fibrosis [17].

In humans, heterozygous mutations of either *ACVRL1* (encoding ALK1) or *ENG* (encoding Endoglin, an endothelial co-receptor for MP9/10) genes cause hereditary hemorrhagic telangiectasia (HHT), a genetic disease with multisystemic vascular defects ranging from small cutaneous and mucosal telangiectasias, to severe arteriovenous malformations (AVMs) in the lung, liver, and the brain, which drive severe cardiac complications [18,19]. HHT presents significant phenotypic variability, wherein the age of symptoms onset, the location of telangiectasias and AVMs, and the severity of the disease vary widely between individuals carrying the same mutation, even within the same family. These significant intra-familial differences support a role for genetic modifiers [20]. Indeed, the PTPN14gene, encoding the non-receptor tyrosine phosphatase 14, was identified as a potential modifier gene whose polymorphisms influence the severity of pulmonary arteriovenous malformations [21]. Heterozygous mice for *Acvrl1* or *Eng* reproduce some HHT-like lesions but with a low frequency [22]. Interestingly, these HHT-like lesions have been shown to be more frequent in the 129/Ola than in the C57BL/6 genetic background, suggesting again that genetic modifiers might play a role in susceptibility to the HHT disease [23,24].

These data prompted us to evaluate the phenotype of *Bmp9*-KO mice under distinct genetic backgrounds, namely C57BL/6, 129/Ola, and BALB/c. We found large differences between these different mouse strains. *Bmp9* deletion leads to premature mortality only in the 129/Ola genetic background. Analysis of the livers of C57BL/6 versus 129/Ola mice in both WT and *Bmp9*-KO animals showed that loss of *Bmp9* in the C57BL/6 strain did not lead to liver fibrosis nor to LSEC capillarization. Our data further suggest differences in the status of the LSEC-differentiated state between these two mouse strains.

## 2. Materials and Methods

### 2.1. Animal Experimentations

Institutional guidelines elaborated by the European Community for the Use of Experimental Animals were followed for all animal experiments (agreement APAFIS#9436-2017032916298306) the approval of the CEA ethics committee and the French Ministry of Research and Education were received. We obtained *Bmp9*-KO mice in the C57BL/6 genetic background from Dr. Se-Jin Lee (Johns Hopkins University, Baltimore, MD). These mice were previously characterized [25] and back-crossed for 10 generations with either 129-P2/OlaHsd wild-type (WT) mice [26] (Harlan/Envigo, Gannat, France) or BALB/c WT mice (Charles River, Ecully, France) in order to obtain *Bmp9*-KO mice from these two genetic backgrounds. The 129-P2/OlaHsd WT mice are hereafter called 129/Ola for the sake of simplification. These mice were bred under a specific-pathogen-free (SPF) animal facility. We rapidly noticed that the *Bmp9* deletion in the 129/Ola background lead to a damageable phenotype with premature mortality—one week before death, these mice presented an important weight loss (10–20% of their initial weight). Subsequently, the mice were weighted once a week and euthanized as soon they had lost 20% of their weight. 

### 2.2. Histological and Immunohistochemical Procedures

Livers were harvested from carbon dioxide-euthanized animals. Organs were fixed by overnight immersion in 4% formaldehyde and then embedded in paraffin after progressive alcoholic dehydration steps. Hematoxylin-eosin and Sirius-red stainings were performed on 5-µm-thick deparaffinized sections (all products from Sigma-Aldrich, St. Louis, MO, USA). The fibrotic response was defined as the ratio between the Sirius-red-positive areas and the total surface of the tissue section. For Collagen IV immunohistochemistry, deparaffinized sections were incubated overnight at room temperature with a primary antibody (Abcam ab19808), after Tris-EDTA buffer antigen retrieval protocol. Appropriate secondary antibody was subsequently incubated for 1 h at room temperature. All microscopy pictures were taken using a Zeiss Axioplan microscope (Zeiss, Oberkochen, Germany) and analyses and quantifications were performed using the Axiovision 4.9.1 image analysis software (Zeiss). Briefly, for the sinusoid area quantification, white areas corresponding to the vessels including veins, sinusoids, and arteries were automatically detected and then veins, arteries and bile ducts were manually removed and the area of the sinusoids were calculated as the percentage of the entire picture area.

### 2.3. Liver Sinusoid Endothelial Cells (LSEC) Isolation and Culture

These cells were isolated as previously described [26]. In brief, we perfused livers from anesthetized mice (26–30-weeks-old females) via the inferior veina cava with warm EGTA buffer (while the portal vein was sectioned), and subsequently by warm collagenase II (Merck-Biochrom, Berlin, Germany). Livers were harvested and mechanically dissociated. Cells were isolated by centrifugation through Histodenz (Sigma-Aldrich) gradients.

Freshly isolated LSECs were immediately used (before cell culture) for mRNA expression studies.

For scanning electron microscopy (SEM), LSECs were seeded on fibronectin-coated coverslips and grown in EGM-2 supplemented with 10% FCS for 4 to 6 h under a 19% O_2_ and 5% CO_2_ atmosphere. This condition was considered as optimal for cell spreading and fenestration observation. 

### 2.4. Scanning Electron Microscopy (SEM)

LSECs were fixed overnight with 2% glutaraldehyde in 0.125 M phosphate buffer pH 7.2. They were then post-fixed for 1 h with 1% osmium tetroxide (OsO_4_), dehydrated with graded ethanol and desiccated by a 30 min immersion in hexamethyldisilazane followed by air drying. Samples were copper-metallized and observed on a Quanta 250 FEG scanning electron microscope at the “Centre Technologique des Microstructures” of University Claude Bernard (Lyon, France). For each condition, fenestrae were quantified on 86 to 105 images, using the multi-point tool of the ImageJ software.

### 2.5. RNA Extraction and Quantitative RT-PCR 

Fresh livers were homogenized in RNA later (Sigma-Aldrich). RNA extraction was carried out from 80–100 mg of tissue using Trizol (Thermo Fisher Scientific, Carlsbad, CA, USA), according to the manufacturer’s instructions. For freshly isolated LSECs, RNA was extracted using the RNA-XS Plus extraction kit (Macherey-Nagel, Düren, Germany). A total of 1 µg of RNA was reverse-transcribed using the iScript kit (Biorad, Hercules, CA, USA). Quantitative RT-PCR experiments were performed using the SYBR-green master-mix GoTaq (Promega, Charbonnières-les-Bains, France) on a CFX96 thermocycler (Biorad). mRNA expression levels were normalized to the level of either VE-cadherin (Cdh5) or Rpl13a, as indicated by the figure legends. All primers are detailed in the Appendix A. The qPCR data were calculated using the ∆∆Ct method or the ∆Ct method as specified in the figure legends.

### 2.6. Enzyme-Linked Immunosorbent Assay (ELISA)

Hyaluronan levels from mouse plasma (40-weeks-old females) were quantified according to the manufacturer’s recommendation (DHYALO Biotechne, Minneapolis, MN, USA).

BMP9 ELISA measurement has been previously described [4]. MAB3209 (R&D Systems) antibody for BMP9 was used as capture antibody and detection was performed using biotinylated antibody (anti mature BMP9: BAF3209, R&D Systems). Plasma of 12 (129/Ola WT) and 12 (C57BL/6) mice were quantified.

### 2.7. BMP Activity Measurement

NIH-3T3 cells were transfected with a mixture of the reporter plasmid pGL3(BRE)2-luc encoding firefly luciferase, downstream of a BMP response element, pRL-TK luc encoding Renilla luciferase and a plasmid encoding human ALK1, as previously described [7].

### 2.8. Statistical Analysis

Data were analyzed using the Prism6 software (GraphPad, La Jolla, CA, USA). The non-parametric tests (Mann–Whitney or Kruskal–Wallis) used are indicated in the figure legends. Results were considered to be statistically significant when *p*-value < 0.05.

## 3. Results

### 3.1. Bmp9 Deletion Leads to Premature Mortality in a Gender- and Strain-Dependent Manner

*Bmp9* genetic deletion was initially generated in C57BL/6 mice by the team of Dr. S.J. Lee. These mice were viable and fertile and exhibited no overt defect under SPF husbandry conditions, except for lymphatic vessels dilation and lymphatic valves malformations [25,27]. To address whether *Bmp9* deletion could lead to different phenotypes depending on the genetic background, these mice were intercrossed for 10 generations with wild-type 129/Ola or BALB/c mice, in order to generate *Bmp9*-KO mice under these two additional genetic backgrounds. We then analyzed their overall survival at the adult stage. In agreement with others and our previous work [25,28], there was no effect of *Bmp9* deletion on the overall survival in the C57BL/6 genetic background (Figure 1A). The overall survival of *Bmp9*-KO mice in the BALB/c strain was also not different from that of WT mice (Figure 1B). In contrast, *Bmp9* deletion in the 129/Ola strain led to premature death (Figure 1C). Interestingly, we observed a clear difference between the males and the females. Males died earlier (mean survival age of 28 weeks) than females (mean survival age of 49 weeks) (Figure 1C). Careful analysis of the different organs of these mice at autopsy revealed macroscopic defects in two main organs—livers had patchy white spots on their surface, as previously described [15] and sometimes hyperdilated vessels. Kidneys also displayed patchy white spots, some were hemorrhagic and sometimes translucid. Kidneys were affected in 40% of the males versus 4% of the females while the liver was affected in both males (22%) and females (45%) (Figure 1D). We noticed that death of males was often preceded by a severe weight-loss (10–20% loss over the week preceding death) (data not shown), suggesting that the defects in the kidney function could be the cause of male death. This early mortality in males explains in part the lower percentage of liver defects in males as they die before developing liver defects. Taken together, these data show a clear difference in the overall survival of *Bmp9*-KO mice, depending on gender and strain.

### 3.2. Bmp9 Deletion in the C57Bl/6 Strain Does not Lead to Liver Fibrosis

We have recently shown that *Bmp9* deletion in the 129/Ola genetic background leads to spontaneous liver fibrosis [15]. We thus wanted to know if *Bmp9* deletion under other genetic backgrounds could also lead to liver fibrosis. We focused on the C57BL/6 genetic background which is the main background studied in gene KO experiments. For this, liver sections of adult WT and *Bmp9*-KO mice from 129/Ola and C57BL/6 strains were stained with Sirius red. We found a significant increase in Sirius-red-stained collagen deposits in 129/Ola *Bmp9*-KO livers (Figure 2A,B) but none in the C57BL/6 *Bmp9*-KO livers. Occasionally, we noted few red spots on the liver sections of the C57BL/6 *Bmp9*-KO mice. We have also previously shown that liver sections from 129/Ola *Bmp9*-KO mice presented dilated sinusoidal vessels [15]. We thus stained liver sections with Hematoxylin-Eosin and measured the surface of sinusoidal vessels in each condition, excluding veins and arteries from this quantification. We found no difference in the sinusoidal size between WT and *Bmp9*-KO mice from the C57BL/6 strain (Figure 2C,D) whereas, the sinusoids were significantly dilated in *Bmp9*-KO livers from the 129/Ola mice, as previously shown [15]. Together, these data showed that *Bmp9* deletion in the C57BL/6 background did not lead to liver fibrosis nor to sinusoidal vessel enlargement.

### 3.3. Bmp9 Deletion in the C57Bl/6 Strain Does Not Modify Liver Sinusoidal Endothelial Cell (LSEC) Differentiation State

We have previously shown that *Bmp9* deletion led to liver fibrosis, which was preceded by capillarization of LSEC [15]. We thus analyzed these different parameters in the LSEC isolated from adult livers of WT versus *Bmp9*-KO mice from the C57BL/6 strain. We first analyzed LSEC fenestration by scanning electron microscopy (SEM) after LSEC plating and culture for 18 h, as previously described [15]. Surprisingly, we found a very small number of fenestrae in the WT LSEC from the C57BL/6 strain (data not shown), compared to the WT LSEC from the 129/Ola strain [15]. As LSEC fenestration decreases rapidly in cells during in vitro culture [29], we looked at an earlier time point that was after 6 h of culture. Again, we observed a very low density of fenestrae (< 0.5 fenestrae/µm^2^; Figure 3A) as compared to the number of fenestrae from the WT LSEC derived from 129/Ola mice under the same culture conditions (>5 fenestrae/µm^2^; Appendix A). When we compared the LSEC fenestration from WT to the *Bmp9*-KO mice in the C57BL/6 strain, we found no significant difference (Figure 3A). Several specific markers of the LSEC terminal differentiation were previously described by Geraud et al. [30], including Gata4 and Maf, two key hepatic transcription factors, Plvap (plasmalemmal vesicle-associated protein; it encodes a constitutive fenestra protein), Stabilin-1 and Stabilin-2 encoding the scavenger receptors, Ehd3 encoding an endocytic receptor, and Cd209b encoding an adhesion molecule. The mRNAs of these different markers were found to be significantly decreased in the 129/Ola *Bmp9*-KO mice, as compared to the WT mice [15]. We thus analyzed the expression of these markers in the C57BL/6 LSECs from the WT and *Bmp9*-KO mice. The expression of these genes was evaluated by RT-qPCR. As shown in Figure 3B, none of these mRNAs was differentially expressed between the *Bmp9*-KO and WT mice. Another marker of endothelial capillarization is the deposition of a basal lamina around sinusoidal endothelial cells, which is normally absent in a normal liver [31]. We thus stained the liver sections from WT and *Bmp9*-KO mice with an antibody against Collagen IV, which is one of the main constituents of basal lamina. We observed a comparable level of Collagen IV staining (Figure 3C) in the WT and the *Bmp9*-KO mice from the C57BL/6 strain. This result was different to the one observed in the 129/Ola background. In this background, the level of collagen IV was very low in the WT mice and high in the *Bmp9*-KO (Appendix A), in accordance with our previous work [15]. We also assessed Collagen4a4 mRNA expression by RT-qPCR and found similar levels between the WT and the *Bmp9*-KO mice (Figure 3D). Finally, we measured circulating hyaluronan levels, which reflected the activity of the scavenger receptor Stabilin-2, in the plasma of these mice. No difference in the HA circulating levels was found between the WT and the *Bmp9*-KO mice (Figure 3E). Taken together, these results showed no difference between the LSEC from the WT and the *Bmp9*-KO in the C57BL/6 background. It clearly appeared, however, that the LSEC from the WT C57BL/6 mice presented a capillarized phenotype, with a low density of fenestrae and the presence of a basal lamina, along the sinusoids. 

### 3.4. C57BL/6 and 129/Ola LSEC from WT Mice Express Different Levels of Hepatic Endothelial Terminal Differentiation Markers

We next compared the LSEC terminal differentiation markers between the WT mice from the C57BL/6 and the 129/Ola strains. Interestingly, we found a significant difference in the mRNA levels of Stabilin-2, Plvap, and Cd209b, which were significantly lower in the C57BL/6 WT mice, as compared to the 129/Ola WT mice (Figure 4). The same trend, although it did not reach statistical significance, could also be observed in the expression of the two transcription factors Gata4 and Maf. On the other hand, the mRNA levels of Stab1, Edh3, and eNOS were not different between the two strains. VE-Cadherin mRNA levels were not different between the two strains, supporting the idea that we analyzed the same amount of LSEC mRNA between the two strains. Taken together, these data indicate again that C57BL/6 LSEC express a less terminally differentiated and a more “capillary-like” phenotype than the 129/Ola LSEC.

### 3.5. C57BL/6 WT Mice Have Lower Levels of Circulating BMP9 and BMP-Activity than 129/Ola Mice

We then investigated the BMP9/ALK1 signaling pathway first by RT-qPCR on freshly prepared LSECS from the C57BL/6 and 129/Ola strains. No difference in the expression of the BMP9 receptors (ALK1, ALK2, ALK5, BMPR2, ActR2A, ENG) could be found between the two strains (Figure 5A). mRNA analysis of the different BMP-specific SMADs (Smad1, Smad5, and Smad9) again showed no difference between the two strains. However, there was a significant decrease in C57BL/6 LSEC versus 129/Ola of Smad6 mRNA levels and a slight decrease of the Id1 mRNA levels, which are the two main BMP9 target genes (Figure 5A). Smad7 mRNA levels were not different between the two strains. We next measured the plasmatic BMP9 levels by ELISA in the two strains. We found a significantly lower level of circulating BMP9 in the WT C57BL/6 mice versus the 129/Ola mice (Figure 5B). This decrease was confirmed when we measured the BMP-ALK1-dependent activity from these plasma, using a BRE-luciferase reporter assay (Figure 5C). 

### 3.6. C57BL/6 WT Mice Express Lower Vegfr2 mRNA Levels than 129/Ola Mice

We then investigated whether the Vascular Endothelial Growth Factor (VEGF) signaling pathway which has been previously described as a key player in the LSEC capillarization [29,32], was different in the liver of WT C57BL/6 mice versus that of WT 129/Ola mice. Vegfa mRNA levels were not different between the two strains (Figure 6). However, we found that Kdr (VEGFR2) mRNA levels were significantly decreased in C57BL/6 as compared to the 129/Ola mice, while the mRNA levels of the other receptors of this pathway Flt1 (VEGFR1) and Flt4 (VEGFR3) were not different between the two strains (Figure 6). 

## 4. Discussion

BMP9 is a growth factor, produced by hepatic stellate cells that is found in the circulation under biological active concentrations [3,4]. It has been reported to act on the vascular endothelium to maintain vessel quiescence via its high affinity endothelial-specific receptor ALK1 [7,9]. We have recently reported that BMP9 plays a central role in liver homeostasis as genetic deletion of *Bmp9* in the 129/Ola mice led to spontaneous hepatic fibrosis [15]. Here, we showed that this effect is strain-dependent, as *Bmp9* deletion in the C57BL/6 strain does not lead to spontaneous liver fibrosis. Comparing the LSEC status in the two different genetic backgrounds showed a differential state of sinusoidal endothelial differentiation. Indeed, C57BL/6-derived LSEC presented several characteristics that suggested a less differentiated and “capillary-like” phenotype, as compared to the 129/Ola LSEC. The already capillary-like phenotype of LSEC derived from the C57BL/6 mice might explain why *Bmp9* deletion, which we previously showed to induce LSEC capillarization [15], did not lead to a more pronounced capillarized phenotype. 

One original observation of the present work is that the phenotypic alterations induced by genetic deletion of *Gdf2/Bmp9* in mice vary as a function of the genetic background of the inbred mouse strain used. We report here that the 129/Ola strain is highly sensitive to *Bmp9* gene deletion, leading to premature death during adulthood. In contrast, C57BL/6 and BALB/c *Bmp9*-KO mice have a normal life expectancy. It has been long known that many biological responses vary from one strain of inbred mice to another and in particular the angiogenic response [33]. In particular, 129/Ola mice have been shown to be more sensitive to vascular stimuli than the C57BL/6 mice [34]. Difference between mouse strains has also been shown to affect genetic regulation of the liver gene expression network [35]. This could result from genomic differences as the deep sequencing of the genomes of 36 commonly used inbred mouse strains by the Wellcome Trust Sanger Institute, recently allowed to identify more than 800 protein-encoding genes with the premature stop codons reducing the length of the proteins by more than 20% [36]. This work, together with our recently published study, clearly supports a role for BMP9 in liver homeostasis, which depends on the genetic background and supports the notion of genetic modifiers in general and more particularly in HHT [20,21]. Our work further supports that studying the function of a protein should not be limited to one mouse line, as one could miss the function of this protein. 

In the present work, we identified major differences in the state of mouse LSEC differentiation between C57BL/6 and 129/Ola strains. Collagen IV staining of liver sections of C57BL/6 mice showed the presence of a basal lamina that was not present in the 129/Ola strain (Figure 3C and Appendix A). Analysis of freshly isolated LSEC from these two strains also showed a lower density of fenestrae in the C57BL/6 mice versus the 129/Ola (Figure 3A and Appendix A). Furthermore, consistent with these results, several markers that have been specifically identified in LSEC [30] were differentially expressed (Stabilin2, Plvap, and Cd209b) between these two mouse strains. Plvap is a dimeric protein constituent of the diaphragm of diaphragmed-fenestrae (present in kidney peritubular capillaries, capillaries of intestinal villi, pancreas, adrenal cortex, endocrine glands, and choriocapillaries of the brain and eye) [37] but it has also been shown to be required for the formation of non-diaphragmed fenestrae, such as those present in LSECs [38]. Interestingly, in line with our results, it has been recently shown by two different groups that deletion of the *Pvlap* gene leads to phenotypes that depend on the genetic background [39,40]. On pure C57BL/6J background, homozygous *Plvap* deletion resulted in 100% embryonic lethality, while in a mixed background some pups were born and survived for up to 4 weeks. *Stab1^-/-^-Stab2^-/-^* mice have been generated in two different backgrounds and it was shown that these double KO mice on a C57BL/6 background showed only moderately elevated levels of albumin in the urine, in comparison to the BALB/c background [41]. These data, together with our results, demonstrated that fenestrae formation and expression of terminal differentiation markers are dependent on the genetic background. Interestingly, we found a significantly lower level of active circulating BMP9 in the plasma of C57BL/6 mice versus that of the 129/Ola mice, which could, at least in part, explain the differences of the differentiation states between these two strains, as we have previously demonstrated that BMP9 is a key factor for LSEC maturation [15]. Interestingly, we also found that livers from the C57BL/6 expressed a lower level of Kdr (VEGFR2) mRNA levels than the 129/Ola mice. Maintenance of the LSEC differentiation was shown to require VEGF-A [29,32]. Thus, a lower level of the VEGFR2, the main VEGF-A receptor, could also explain the “capillarized” state of the WT LSEC from the C57BL/6 versus that of the 129/Ola mice. 

Our data suggest a different state of the LSEC differentiation between these two mouse strains. C57BL/6 LSEC seem to have a more “capillary-like” phenotype than the 129/Ola LSEC. Capillarization has been reported to precede liver fibrosis in human patients and animal models [17,31], thus, one would expect to see more spontaneous liver fibrosis in the C57BL/6 mice. However, we did not observe liver fibrosis in the WT C57BL/6 mice. It is, thus, likely that the C57BL/6 mice have developed a physiological adaptation to this peculiar LSEC phenotype. In particular, since LSEC fenestrae appear to facilitate the uptake of chylomicron remnants from the portal circulation, it remains to be understood whether the remaining number of fenestrae is sufficient for lipid uptake or if C57BL/6 mice have developed alternative mechanisms to control lipidemia.

The absence of spontaneous liver fibrosis in *Bmp9*-KO mice in the C57BL/6 background is compatible with the work of Breitkopf-Heinlein et al., [3]. In their work, they proposed that BMP9 stabilizes hepatocyte function in healthy livers but that BMP9 could become harmful under conditions of liver damage. These opposites functions of BMP9 have also been observed in malignant versus non-malignant liver conditions [42,43], supporting the context-dependent role of BMP9. It will be interesting in the future to challenge *Bmp9*-KO mice of the 129/Ola background, which spontaneously develop liver fibrosis under conditions of liver damage. In the present work, we have only addressed the role of BMP9 on LSEC, but it has been demonstrated in previous studies that BMP9 can modulate HSC and hepatocyte activities [3,11]. It will, thus, be important to study the finely tuned crosstalk between the different liver cell types driven by BMP9 to better understand the role of this key factor in liver homeostasis.

The more differentiated LSEC phenotype observed in the 129/Ola mice might explain why this mouse strain is more prone to liver fibrosis in the absence of *Bmp9*. This could in part explain, why it is so difficult to develop liver fibrosis models in mice as most studies are performed in C57BL/6 mice [44]. Together, our results suggest that the 129/Ola strain seems to be a good model to study liver fibrosis mediated by capillarization. It has been clearly demonstrated that capillarization precedes the onset of alcoholic liver disease in humans [45]. It would, thus, be interesting in the future to challenge the *Bmp9*-KO mice of the 129/Ola genetic background with a high-fat diet or to test whether these mice could develop cirrhosis and hepatocarcinoma. 

## Figures and Tables

**Figure 1 cells-08-01079-f001:**
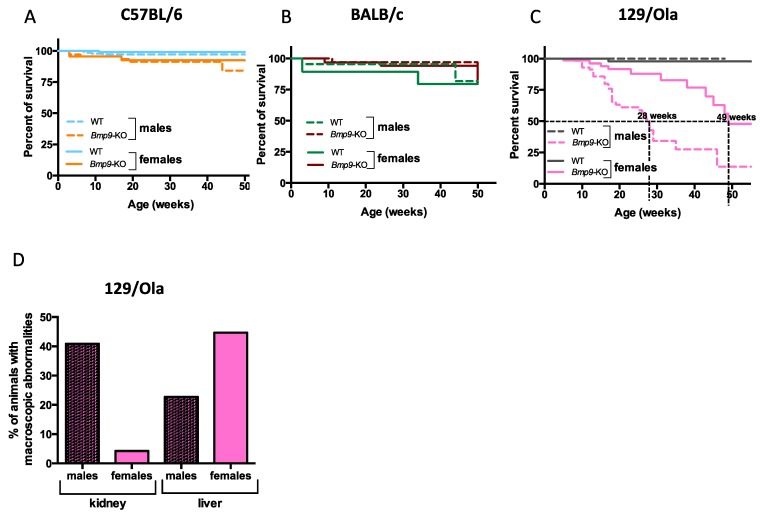
Survival curves of *Bmp9*-KO mice of three genetic backgrounds. (**A**) Survival in C57BL/6 background (wild-type (WT) males *n* = 154, WT females *n* = 168, *Bmp9*-KO males *n* = 63, *Bmp9*-KO females *n* = 43). (**B**) Survival in BALB/c background (WT males *n* = 42, WT females *n* = 49, *Bmp9*-KO males *n* = 44, *Bmp9*-KO females *n* = 75). (**C**) Survival in 129/Ola background (WT males *n* = 62, WT females *n* = 61, *Bmp9*-KO males *n* = 44, *Bmp9*-KO females *n* = 59). (**D**) Macroscopic alterations of kidneys and the liver revealed at autopsy in 129/Ola *Bmp9*-KO mice before 50 weeks of age. Results are expressed in terms of percentage of total autopsied animals (females *n* = 51; males *n* = 61).

**Figure 2 cells-08-01079-f002:**
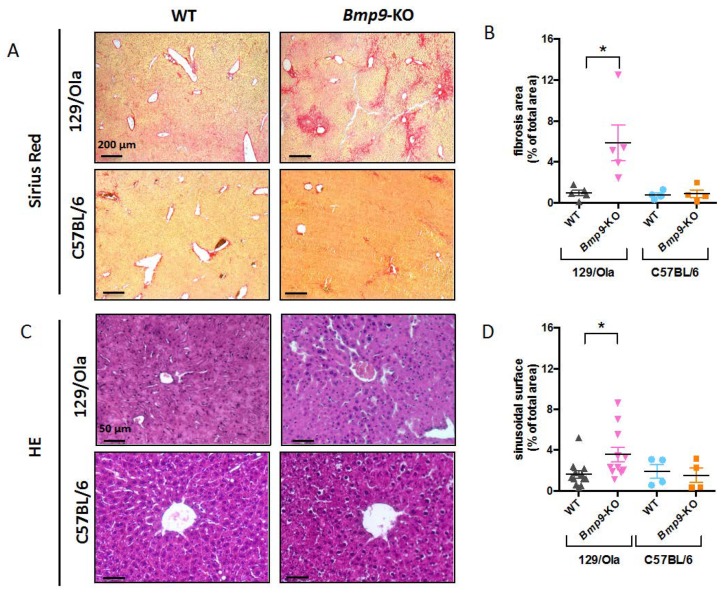
C57Bl/6 and 129/Ola *Bmp9*-KO mice display different liver phenotypes. (**A**) Sirius red histochemical staining of the liver sections (5 µm) of WT and *Bmp9*-KO mice from the 129/Ola and C57BL/6 strain. (**B**) Quantification of fibrotic surface in percentage of total area (*n* = 4 in each group, 5 images analyzed per mouse). Fibrotic areas appear in red and were quantified using the Axiovision software. (**C**) Hematoxylin and Eosin histochemical staining of liver sections (5 µm) from WT and *Bmp9*-KO mice from 129/Ola and C57BL/6 strain. (**D**) Quantification of sinusoidal surface measured using Axiovision software and expressed as percentage of total area (*n* = 4–12 mice per group, 5 images analyzed per mouse). Central veins and arteries were excluded from this quantification. Data are expressed as mean ± standard error of the mean (SEM) of duplicate determinations from multiple samples. The Kruskal–Wallis test was used for all statistical analysis of this figure. **p* < 0.05.

**Figure 3 cells-08-01079-f003:**
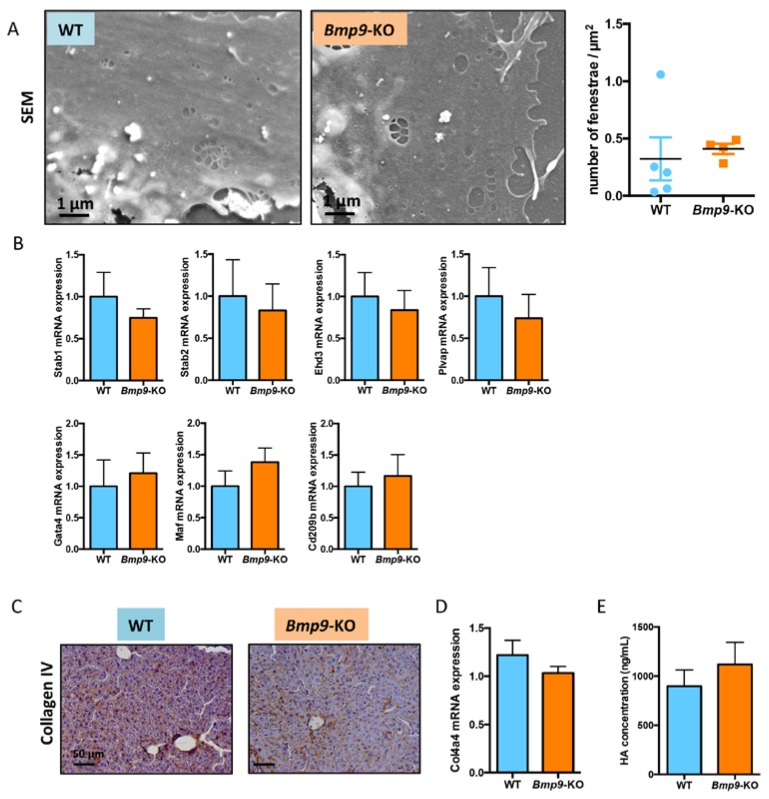
Liver sinusoidal endothelial cells (LSEC) capillarization in WT and *Bmp9*-KO mice in the C57BL/6 genetic background. (**A**) Representative pictures of Scanning Electron Microscopy (SEM) on freshly isolated LSEC from C57BL/6 mice (26-weeks-old females), 4–6 h after plating. Graph on the right shows the quantification of fenestrae per µm^2^ assessed using the ImageJ software (*n* = 5 for WT mice, 86 images quantified; and *n* = 4 for *Bmp9*-KO mice, 105 images quantified). (**B**) mRNA expression levels of the LSEC markers measured by the RT-qPCR (*n* = 5 and 4 for respectively WT and *Bmp9*-KO mice). mRNA expression levels were normalized to Vascular Endothelial Cadherin (VE-Cadherin) (Cdh5). The results are expressed as a ratio to the WT mean value (∆∆Ct method). (**C**) Representative images of Collagen IV immunohistochemical staining on liver sections (5 µm) from WT and *Bmp9*-KO mice in the C57BL/6 genetic background (*n* = 4 in each group). (**D**) mRNA expression of Col4a4 in livers from WT and *Bmp9*-KO mice in the C57BL/6 genetic background (*n* = 6 in each group). Values were normalized to Rpl13a mRNA expression levels. The results are expressed as a ratio to the WT mean value. (**E**) Circulating levels of hyaluronan (HA) were measured by ELISA in at least 6 plasma samples from both WT and *Bmp9*-KO mice of the C57Bl/6 genetic background. Data are plotted as mean ± (SEM). All data were statically analyzed with the Mann–Whitney test. No significant difference was observed (*p*-values > 0.05).

**Figure 4 cells-08-01079-f004:**
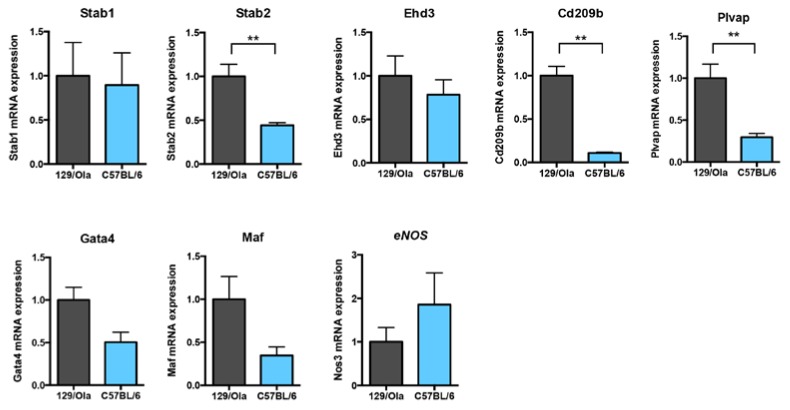
mRNA levels of LSEC differentiation markers in the WT mice of 129/Ola and C57BL/6 genetic backgrounds. mRNA expression levels of the LSEC markers were analyzed by RT-qPCR from freshly prepared LSEC from six 129/Ola and four C57BL/6 WT mice. Values were normalized to Rpl13a mRNA expression levels and the results are expressed as a ratio of these values to that of 129/Ola (∆∆Ct method). Data are plotted as mean ± SEM. Statistical analysis was performed using the Mann–Whitney test. ** *p* <0.01.

**Figure 5 cells-08-01079-f005:**
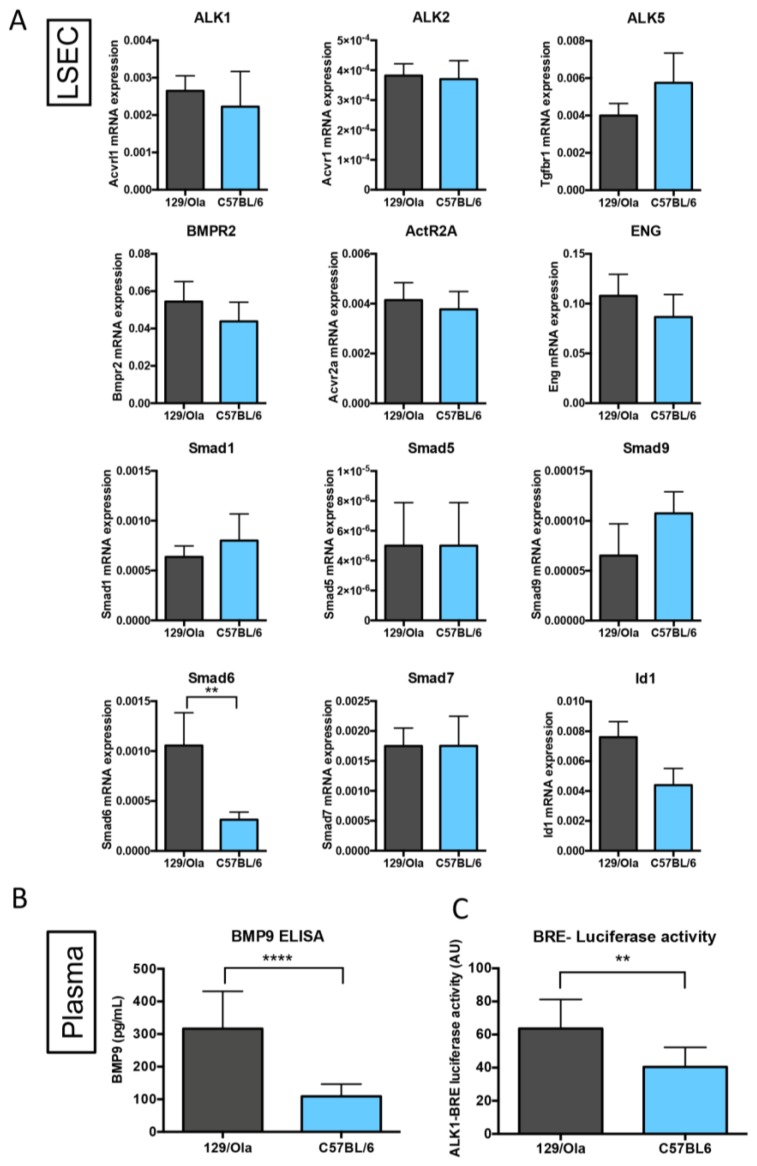
BMP9/ALK1 signaling pathway in 129/Ola and the C57BL/6 WT mice. (**A**) Quantification of the mRNA expression level of type I (ALK1, ALK2, ALK5) and type II (BMPR2, ActR2A) receptors, the coreceptor endoglin, the transcription factors (Smad1, Smad5 and Smad9), and the target genes (Smad6, Smad7 and Id1), from freshly prepared LSEC of six 129/Ola and four C57BL/6 mice. Expression was normalized to the Rpl13a mRNA expression (∆Ct method). (**B**) BMP9 plasma levels measured by ELISA in the 129/Ola and the C57BL/6 WT mice (males, 6 to 8 months old, *n* = 12 in each group). (**C**) ALK1-BRE luciferase activity (AU: Arbitrary Units) in plasma from 129/Ola and C57BL/6 WT mice (6–8-months-old males, *n* = 12 in each group). Data are plotted as mean ± SEM. Statistical analysis was performed using the Mann–Whitney test. ** *p* < 0.01; **** *p* < 0.0001.

**Figure 6 cells-08-01079-f006:**
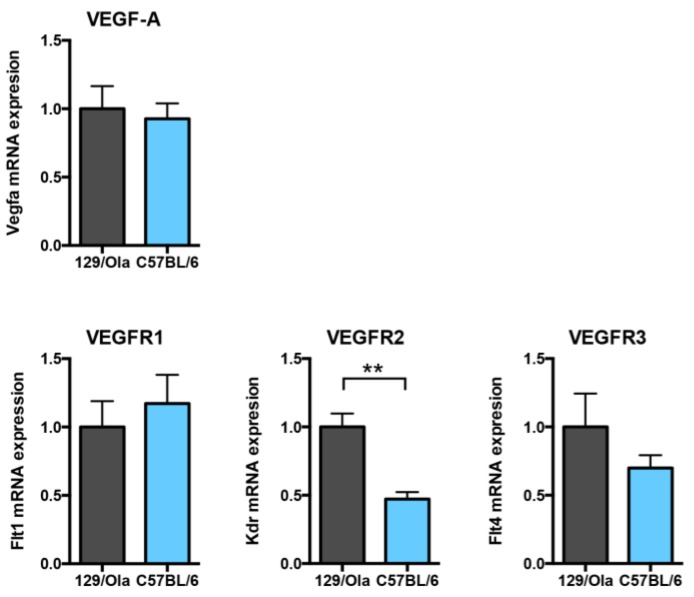
mRNA levels of the Vascular Endothelial Growth Factor (VEGF) signaling pathway in the livers of WT mice of 129/Ola and C57BL/6 genetic backgrounds. Quantification of the mRNA expression level of Vegfa (VEGF-A) and its three receptors Vegfr2, Flt1 (VEGFR1), and Flt4 (VEGFR3) from the livers of the 129/Ola and the C57BL/6 mice (*n* = 4–5 and *n* = 6, respectively). Values are normalized to the Rpl13a mRNA expression levels and the results are expressed as a ratio of these values to that of 129/Ola (∆∆Ct method). Data are plotted as mean ± SEM. Statistical analysis was performed using the Mann–Whitney test. ** *p* < 0.01.

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
