# Peer review of "Differential Consequences of Bmp9 Deletion on Sinusoidal Endothelial Cell Differentiation and Liver Fibrosis in 129/Ola and C57BL/6 Mice"

_cells, 2019, doi:10.3390/cells8091079_

Round 1

Reviewer 1 Report

This very interesting study by Desroches-Castan and colleagues reports about a possible function of Bone Morphogenetic Protein (BMP)-9 in the homeostasis of mouse liver sinusoidal endothelial cells (LSEC).

As the authors correctly mention, the differentiation state of the LSEC is an important criterion because dedifferentiation (=capillarization) is believed to represent an early sign of liver damage and it typically precedes the development of liver fibrosis. Therefore better knowledge about the regulation mechanisms of LSEC capillarization should help to better understand the mechanisms of liver homeostasis as well as disease.

The authors have previously published that deletion of BMP-9 in 129/Ola mice results in spontaneous formation of liver fibrosis, including LSEC capillarization. These data are in contrast to results that were obtained with BMP-9 KO mice on a C57BL/6 genetic background, which did not show any spontaneous development of liver damage. Even somehow in contrast, in these Black-6 mice, when challenged with CCl4, absence or neutralization of BMP-9 was protective (Breitkopf-Heinlein K., et al., Gut 2017). The surprising conclusion from these opposing observations would be that in Black-6 mice BMP-9 acts pro-fibrogenic whereas in 129/Ola mice it seems to protect LSEC from capillarization, which is an anti-fibrotic feature.

Up to now there is no good explanation for these different responses in the two mouse strains. Therefore in the present work the authors directly compared the livers and the isolated LSECs from the two mouse strains. They convincingly demonstrate that wild-type LSEC from Black-6 mice are less differentiated than LSEC from 129/Ola mice. Interestingly especially the male 129/Ola KO mice not only manifested with increasing liver damage but also with some kidney malformations, both of which were not observed in C57BL/6 or BALB/c KO mice. In addition, presence and signalling of BMP-9 seems generally higher in 129/Ola mice than in Black-6 and the livers of 129/Ola mice express more VEGF-Receptor (Kdr) mRNA than Black-6.

In general it is of great importance to know how exactly BMP-9 acts on LSEC, making these new data very relevant. However, the conclusions that can be drawn from these findings still need some discussion (see below). And the question about the role that BMP-9 finally plays in humans will need further research.

Detailed comments/criticism:

Title: “BMP9 role in…”; this is not correct, either “BMP9’s role” or “the role of BMP9 in…” Abstract (and results, line 308-310; discussion, lines 388-389 and 392-393): “VEGFA and BMP9 are two key players of LSEC differentiation and thus their low levels could explain in part the capillarized phenotype of WT C57BL/6 LSEC” – This conclusions is a bit exaggerated: if BMP-9 really protects LSEC from becoming capillarized, why do then the C57BL/6 KO not develop any (further) spontaneous damage? In contrast, when challenged with CCl4 the KO’s are even more resistant to damage. This does not fit with the hypothesis of BMP-9 being a general antagonist of capillarization. Introduction line 36: “…was first reported to be an autocrine/paracrine cytokine inducing hepatocyte proliferation [5].” – this is not right and should be corrected: BMP-9 was reported to act anti-proliferative on adult (non-malignant) hepatocytes (Breitkopf-Heinlein K., et al., Gut 2017) and in the cited paper 5 (Miller et al.) only data about non-parenchymal cells are described. Results: line 167: please define “macroscopic defects” in livers and kidneys – what exactly was observed? any explanation why only the males had also kidney problems? What is known about the role of BMP-9 in the kidney? Females had a bit more liver problems than males but still they survived better than the males – should be discussed. In general these gender differences are an interesting observation. Figure 4: how about the expression levels of Lyve-1, fibronectin and Col4a4? Discussion: line 351: “This dedifferentiated state could explain the absence of effect of Bmp9 deletion which we previously showed to induce LSEC capillarization [16].” – this sentence is a bit difficult to understand: absence of BMP-9 in C57BL/6 mice is not really without effect, the mice do not spontaneously develop any phenotype but when challenged (e.g. with CCl4) they develop less damage. The present data cannot really explain this, right? line 365: there is one sudden sentence about PTPN14 – is the message that the authors are assuming that polymorphisms in this gene might be involved in the opposing BMP-9 responses of the two mouse strains? How are the polymorphism status, is that known? It is very interesting that the genetic background alone plays such a fundamental role and we should indeed try to identify the exact individual differences that are responsible. If we know these we could try to translate these findings to the clinic – are there individual genetic signatures/polymorphisms making one patient more prone to develop liver disease in combination with BMP-9 than another? This would be very important to know. lines 397-401: the authors suggest that C57BL/6 mice might have somehow adapted to the basal capillarization state of their LSEC and that due to this they do not develop (more) fibrosis when BMP-9 is deleted. If this is the case, it should be the same for the Balb-c mice? And the comment about a possible inability of the Black-6 mice to uptake chylomicron remnants: this has not been investigated yet, right? May be the remaining number of fenestrae is sufficient? lines 406-407: “The LSEC differentiated state found in 129/Ola mice is closer to the LSEC differentiation state found in humans and as thus, this mouse strain seems to be a better model to study liver fibrosis.” – this cannot be concluded. Are humans genetically more similar to 129/Ola mice than to C57BL/6? The Black-6 mice are completely healthy whereas the 129/Ola seem to have some general angiogenic vulnerability – is that right? It remains to be investigated which of the two strains (if any) is the model that is closer to the human situation. In any way: both models show that in general BMP-9 is an important factor in regulating the LSEC differentiation status. In general it might be necessary to differentiate between direct effects of BMP-9 on LSEC capillarization and on “real” fibrogenesis in vivo. The latter process involves a fine-tuned cross-talk between several liver cell types including hepatic stellate cells, LSEC and hepatocytes. How does absence of BMP-9 lead to reduced fibrosis in the CCl4 model in C57BL/6 mice? Does BMP-9 somehow directly activate stellate cells? How does presence or absence of BMP-9 feed back on hepatocytes acute phase responses? And finally how does BMP-9 act on Kupffer cells/macrophages and the immune response in general? It seems that regarding the detailed actions of BMP-9 in the liver there is still a lot of unresolved questions. Therefore it might be better to phrase some of the statements/conclusions a bit more carefully, trying to not over-interpret the individual findings? Additional experiments that could be helpful: isolate LSEC from C57BL/6 and stimulate ex vivo with BMP-9: does this reduce or enhance the basal differentiation state of the cells? If indeed the basal lower level of BMP-9 is already sufficient to induce some degree of capillarization, then addition of (more) BMP-9 should antagonize this. isolate LSEC from 129/Ola mice and culture them +/- BMP-9 inhibition (e.g. addition of Alk1-Fc to the culture medium) – does this lead to enhanced capillarization? Or does the KO-phenotype of these mice perhaps need a more complex interaction/cellular cross-talk?

Author Response

Title: “BMP9 role in…”; this is not correct, either “BMP9’s role” or “the role of

BMP9 in…”

There was also a comment from the second reviewer on the title, so the title has been changed accordingly:“Differential consequences of Bmp9deletion on sinusoidal endothelial cell differentiation and liver fibrosis between 129/Ola and C57BL/6 mice”

Abstract (and results, line 308-310; discussion, lines 388-389 and

392-393): “VEGFA and BMP9 are two key players of LSEC differentiation and

thus their low levels could explain in part the capillarized phenotype of WT

C57BL/6 LSEC” – This conclusions is a bit exaggerated: if BMP-9 really

protects LSEC from becoming capillarized, why do then the C57BL/6 KO not

develop any (further) spontaneous damage?

We agree that this statement is more a working hypothesis than a conclusion and has thus been removed from the abstract and the result section and only kept in the Discussion section. The complexity of BMP9’s role in liver fibrosis is also more discussed and in particular in the context of the work of Breitkopf-Heinlein K., et al (Gut, 2017).

Introduction

line 36: “…was first reported to be an autocrine/paracrine cytokine inducing

hepatocyte proliferation [5].” – this is not right and should be corrected: BMP-9

was reported to act anti-proliferative on adult (non-malignant) hepatocytes

(Breitkopf-Heinlein K., et al., Gut 2017) and in the cited paper 5 (Miller et al.)

only data about non-parenchymal cells are described.

Thanks for this remark. This is indeed an error from us, the sentence and reference have been changed: “BMP9 was reported to be an autocrine/paracrine cytokine inhibiting adult (non-malignant) hepatocyte proliferation (Breitkopf-Heinlein, Meyer et al. 2017) ».

Results: line 167: please

define “macroscopic defects” in livers and kidneys – what exactly was

observed? any explanation why only the males had also kidney problems?

What is known about the role of BMP-9 in the kidney? Females had a bit more

liver problems than males but still they survived better than the males – should

be discussed. In general these gender differences are an interesting

observation.

“Macroscopic defects” is indeed a not very informative term. We now precise in the text what we mean by macroscopic defects: “Livers had patchy white spots on their surface, as previously described (Desroches-Castan, Tillet et al. 2019)and sometimes hyperdilated vessels while kidneys displayed also patchy white spots, were haemorrhagic and sometimes translucid.”

We do not have any explanation why only males had kidney defects. To our knowledge, only one publication talks about BMP9 and kidney (Zhu J Cell Mol Med 2015). In this work, BMP9 was markedly elevated in serum with chronic kidney disease and BMP9 increases alkaline phosphatase activity and calcification. This work does not really make sense with our results. Our data show that females have more liver problems than males but this is due to the fact that males have more kidney defects and die earlier. This is before we can detect liver defects that increase with age as we previously published (Desroches-Castan, Tillet et al. 2019).  If we analyze the % of liver defects in surviving males and females at the same age (between 25-35 week-old or 35-50 week-old, i.e. after the massive death of males), then we observe similar numbers (figure for the reviewer). This point is now discussed in the manuscript:“This early mortality in males explains in part the lower percentage of liver defects in males as they die before developing liver defects.”

We agree with the reviewer that the gender specificity is really interesting and understudied in most works.

-Figure 4: how about the expression levels of Lyve-1, fibronectin

and Col4a4?

These genes are indeed interesting to study in this context. We have analyzed Lyve-1 and fibronectin expression in LSEC derived from 129/Ola and C57BL/6 mice and found no significant difference (see figure below for the reviewer) so we decided not to show them. We have also tried to measure Col4A4 expression in LSEC but the detected levels were too low to be confident with the data and were not analyzed.

Discussion: line 351: “This dedifferentiated state could explain

the absence of effect of Bmp9 deletion which we previously showed to induce

LSEC capillarization [16].” – this sentence is a bit difficult to understand:

This sentence has been changed and we hope that our message is now easier to understand: “The already capillary-like phenotype of LSEC derived from C57BL/6 mice might explain why Bmp9deletion, which we previously showed to induce LSEC capillarization (Desroches-Castan, Tillet et al. 2019), does not lead to a more pronounced capillarized phenotype. “

absence of BMP-9 in C57BL/6 mice is not really without effect, the mice do not

spontaneously develop any phenotype but when challenged (e.g. with CCl4)

they develop less damage. The present data cannot really explain this, right?

No. Our data are compatible with this work (Breitkopf-Heinlein, Meyer et al. 2017)as Bmp9 deletion in C57BL/6 mice does not lead to spontaneous liver fibrosis. However, our data do not explain how the loss of BMP9 in C57BL/6 mice when challenged to CCl4 protects from liver fibrosis.CCl4-induced fibrosis results in altered lipid metabolism and lipid peroxidation and the destruction of polyunsaturated fatty acids that will generate generalized hepatic damage and an increase in inflammation (Sheweita, El-Gabar et al. 2001). The molecular processes leading to fibrosis in response to CCl4 are thus probably very different to the ones occurring here, i.e. perivascular fibrosis probably due to a defect in LSEC differentiation.  

line 365: there is one sudden sentence about PTPN14 – is the message that

the authors are assuming that polymorphisms in this gene might be involved in

the opposing BMP-9 responses of the two mouse strains?

No. This sentence was probably not clear and has be placed in the introduction section:“Indeed, the PTPN14gene, encoding the non-receptor tyrosine phosphatase 14, was identified as one potential modifier gene whose polymorphisms influence the severity of pulmonary arteriovenous malformations (Benzinou, Clermont et al. 2012) ».

 -How are the

polymorphism status, is that known?

? Can talk about genome, polymorphism in mouse?

It is very interesting that the genetic

background alone plays such a fundamental role and we should indeed try to

identify the exact individual differences that are responsible. If we know these

we could try to translate these findings to the clinic – are there individual

genetic signatures/polymorphisms making one patient more prone to develop

liver disease in combination with BMP-9 than another? This would be very

important to know.

The polymorphism status between these two strains has been addressed by others by sequencing the genome of these mice as discussed in the manuscript: “The Wellcome Trust Sanger Institute recently allowed to identify more than 800 protein-encoding genes with premature stop codons reducing the length of the proteins by more than 20% (Steeland, Timmermans et al. 2016)”.

We agree that genetic background is probably very important and this is the message that we want to illustrate by publishing this work. Many other publications have shown similar differences for other genes in other genetic backgrounds. So far, there is no polymorphic signature known in BMP9 signaling that could explain differences between patients but we hope that in the future we will be able to identify these polymorphisms.

lines 397-401: the authors suggest that C57BL/6 mice might

have somehow adapted to the basal capillarization state of their LSEC and that

due to this they do not develop (more) fibrosis when BMP-9 is deleted. If this is

the case, it should be the same for the Balb-c mice?

Yes probably, but we have not studied liver fibrosis or LSEC fenestration in BALB/c mice. So, at this stage, we cannot say more.

And the comment about a

possible inability of the Black-6 mice to uptake chylomicron remnants: this has

not been investigated yet, right? May be the remaining number of fenestrae is

sufficient?

 No, we have not studied this point, this is just a working hypothesis. It is also possible that the remaining number of fenestrae is sufficient. This point has been added to the manuscript:”In particular, since LSEC fenestrae appear to facilitate the uptake of chylomicron remnants from the portal circulation, it remains to be understood whether the remaining number of fenestrae is sufficient for lipid uptake or if C57BL/6 mice have developed alternative mechanisms to control lipidemia. »

lines 406-407: “The LSEC differentiated state found in 129/Ola mice

is closer to the LSEC differentiation state found in humans and as thus, this

mouse strain seems to be a better model to study liver fibrosis.” – this cannot

be concluded. Are humans genetically more similar to 129/Ola mice than to

C57BL/6? The Black-6 mice are completely healthy whereas the 129/Ola seem

to have some general angiogenic vulnerability – is that right? It remains to be

investigated which of the two strains (if any) is the model that is closer to the

human situation.

We agree that this statement is a bit too strong. This sentence has been changed: “Together, our results suggest that the 129/Ola strain seems to be a good model to study liver fibrosis mediated by capillarization.

It is difficult to say which mouse is healthier and which one is more vulnerable, but from the literature it seems clear that129/Ola mice are more responsive to vascular stimuli(Rohan, Fernandez et al. 2000, Chan, Pham et al. 2005)this point is now clarified:”It is long known that many biological responses vary from one strain of inbred mice to another and in particular the angiogenic response (Korshunov and Berk 2004), in particular, 129/Ola mice have been shown to be more sensitive to vascular stimuli than C57BL/6 mice (Rohan, Fernandez et al. 2000), but this was also in the genetic regulation of liver gene expression network (Gatti, Maki et al. 2007). »

In any way: both models show that in general BMP-9 is an

important factor in regulating the LSEC differentiation status. In general it might

be necessary to differentiate between direct effects of BMP-9 on LSEC

capillarization and on “real” fibrogenesis in vivo. The latter process involves a

fine-tuned cross-talk between several liver cell types including hepatic stellate

cells, LSEC and hepatocytes. How does absence of BMP-9 lead to reduced

fibrosis in the CCl4 model in C57BL/6 mice? Does BMP-9 somehow directly

activate stellate cells? How does presence or absence of BMP-9 feed back on The hepatocytes acute phase responses? And finally how does BMP-9 act on

Kupffer cells/macrophages and the immune response in general? It seems that

regarding the detailed actions of BMP-9 in the liver there is still a lot of

unresolved questions. Therefore it might be better to phrase some of the

statements/conclusions a bit more carefully, trying to not over-interpret the

individual findings? Additional experiments that could be helpful: isolate LSEC

from C57BL/6 and stimulate ex vivo with BMP-9: does this reduce or enhance

the basal differentiation state of the cells? If indeed the basal lower level of

BMP-9 is already sufficient to induce some degree of capillarization, then

addition of (more) BMP-9 should antagonize this. isolate LSEC from 129/Ola

mice and culture them +/- BMP-9 inhibition (e.g. addition of Alk1-Fc to the

culture medium) – does this lead to enhanced capillarization? Or does the KOphenotype these mice perhaps need a more complex interaction/cellular

cross-talk?

We agree that the present work and recent publications clearly indicate that BMP9 is an important factor in regulating liver fibrosis even if all the mechanisms are not fully understood and that further work is needed to address these points. We probably did not discuss enough our results in the context of the recently published work from (Breitkopf-Heinlein, Meyer et al. 2017). We have now added a paragraph trying to put into perspectives our data with this work and the other cellular subtypes in the liver:”

The absence of spontaneous liver fibrosis in Bmp9-KO mice in the C57BL/6 background is compatible with the work of Breitkopf-Heinlein and coll.(Breitkopf-Heinlein, Meyer et al. 2017). In their work, they propose that BMP9 stabilize hepatocyte function in healthy liver but that BMP9 can become harmful under conditions of liver damage. These opposites functions of BMP9 have also been observed in liver malignant versus non-malignant conditions (Herrera, Garcia-Alvaro et al. 2013, Li, Gu et al. 2013)supporting the context-dependent role of BMP9. It will be interesting in the future to challenge Bmp9-KO mice in the 129/Ola background that spontaneously develop liver fibrosis under conditions of liver damage. In the present work, we have only addressed the role of BMP9 on LSEC, but it has been demonstrated in previous works that BMP9 can modulate HSC and hepatocytes activities (Herrera, Dooley et al. 2014, Breitkopf-Heinlein, Meyer et al. 2017). It will thus be important to study the finely tuned crosstalk of BMP9 between the different liver cell types driven by BMP9 to better understand the role of this key factor in liver homeostasis. ”

Additional experiments that could be helpful: isolate LSEC

from C57BL/6 and stimulate ex vivo with BMP-9: does this reduce or enhance

the basal differentiation state of the cells? If indeed the basal lower level of

BMP-9 is already sufficient to induce some degree of capillarization, then

addition of (more) BMP-9 should antagonize this.

We have tried in our previous paper to add BMP9 to LSEC derived from Bmp9-KO 129/Ola mice that were already capillarized and we were not able to induce the formation of new fenestrae in vivo (Desroches-Castan, Tillet et al. 2019). What we found is that BMP9 can delay the loss of fenestration but not induce new fenestrae. We believe that formation of fenestrae is a complex process that might not be possible to recapitulate in vitro. BMP9 might also not be the only factor necessary for this process.

isolate LSEC from 129/Ola

mice and culture them +/- BMP-9 inhibition (e.g. addition of Alk1-Fc to the

culture medium) – does this lead to enhanced capillarization? Or does the KOphenotype these mice perhaps need a more complex interaction/cellular

cross-talk?

This is an interesting experiment that we have not done. However, we are not sure that it will give an interesting result as these cells are cultured in absence of BMP9 and BMP9 is not produced by LSEC. We believe that we would need a more complex cellular crosstalk. It could be interesting in the future to try this experiment together with LSEC and HSC.

Breitkopf-Heinlein, K., C. Meyer, C. Konig, H. Gaitantzi, A. Addante, M. Thomas, E. Wiercinska, C. Cai, Q. Li, F. Wan, C. Hellerbrand, N. A. Valous, M. Hahnel, C. Ehlting, J. G. Bode, S. Muller-Bohl, U. Klingmuller, J. Altenoder, I. Ilkavets, M. J. Goumans, L. J. Hawinkels, S. J. Lee, M. Wieland, C. Mogler, M. P. Ebert, B. Herrera, H. Augustin, A. Sanchez, S. Dooley and P. Ten Dijke (2017). "BMP-9 interferes with liver regeneration and promotes liver fibrosis." Gut66(5): 939-954.

Chan, C. K., L. N. Pham, J. Zhou, C. Spee, S. J. Ryan and D. R. Hinton (2005). "Differential expression of pro- and antiangiogenic factors in mouse strain-dependent hypoxia-induced retinal neovascularization." Lab Invest85(6): 721-733.

Desroches-Castan, A., E. Tillet, N. Ricard, M. Ouarne, C. Mallet, L. Belmudes, Y. Coute, O. Boillot, J. Y. Scoazec, S. Bailly and J. J. Feige (2019). "Bone Morphogenetic Protein 9 Is a Paracrine Factor Controlling Liver Sinusoidal Endothelial Cell Fenestration and Protecting Against Hepatic Fibrosis." Hepatology.

Rohan, R. M., A. Fernandez, T. Udagawa, J. Yuan and R. J. D'Amato (2000). "Genetic heterogeneity of angiogenesis in mice." Faseb J14(7): 871-876.

Sheweita, S. A., M. A. El-Gabar and M. Bastawy (2001). "Carbon tetrachloride changes the activity of cytochrome P450 system in the liver of male rats: role of antioxidants." Toxicology169(2): 83-92.

Zhu, D., N. C. Mackenzie, C. M. Shanahan, R. C. Shroff, C. Farquharson and V. E. MacRae (2015). "BMP-9 regulates the osteoblastic differentiation and calcification of vascular smooth muscle cells through an ALK1 mediated pathway." J Cell Mol Med19(1): 165-174.

Reviewer 2 Report

Overall, I thought this was an excellent paper.  I believe the conclusions support the data and the discussion was very well written.  Please consider the following suggestions:

The title could be more specific. Consider adding C57BL/6 and 129/Ola instead of referring only to "genetic background" and discuss BMP9 deletion instead of only broadly, its "role".

Within the results there are sections that may have been better suited for the introduction, such as lines 220-222.

BALB/c is often mentioned, but data is not presented.  If data was generated for BALB/c it could be included (at least in supplement), but I don't believe it is appropriate to discuss BALB/c without any support.

Could the use of CO2 for euthanasia have detrimental affects on the analysis of liver tissue? You may consider other methods to decrease potential artifacts observed in the liver.

A more comprehensive analysis of the genetic background after 10 crosses would put more faith in your results between mouse strains compared.

Lines 226 and 227 discuss time points used, but I was confused as to the rational and picking of the time points.

The error bars in Fig. 3 are quite large and significance cannot be determined.  Consider repeating experiments.

Author Response

The title could be more specific. Consider adding C57BL/6 and 129/Ola

instead of referring only to "genetic background" and discuss BMP9 deletion instead of only broadly, its "role".

The title has been changed: “Differential consequences of Bmp9deletion on sinusoidal endothelial cell differentiation and liver fibrosis between 129/Ola and C57BL/6 mice”

Within the results there are sections that may have been better suited for the introduction, such as lines 220-222.

We agree, and these lines have been switched to the introduction section.

BALB/c is often mentioned, but data is not presented. If data was generated for BALB/c it could be included (at least in supplement), but I don't believe it is appropriate to discuss BALB/c without any support.

We only show BALB/c survival curves and only discuss BALB/c in line with this result. We think that it is important to show this data in comparison to the other two genetic backgrounds as it further supports this notion of genetic variability.

Could the use of CO2 for euthanasia have detrimental affects on the analysis of liver tissue? You may consider other methods to decrease potential artifacts observed in the liver.

CO2 is only used for mouse euthanasia in the experiments studying the whole liver for Histological and immunohistochemical procedures. When we are working with isolated LSEC, animals die of exsanguination. We are not aware of detrimental effects of C02 on liver tissue and if this was the case, WT and Bmp9-KO animals of the different genetic backgrounds were all euthanized identically when compared so this should not strongly affect our conclusions.

A more comprehensive analysis of the genetic background after 10 crosses would put more faith in your results between mouse strains compared.

This is an important point that we all have to be aware of in order to make sure that the phenotype observed corresponds indeed to the deleted gene. We did not sequence these mice before and after the crosses. However, in order to limit the genetic drift, our mice were refreshed with mice from our commercial providers every 5-10 generations.

Lines 226 and 227 discuss time points used, but I was confused as to the rational and picking of the time points.

It is well established that LSEC rapidly lose their healthy quiescent phenotype (fenestrae grouped into sieve plates) during culture (DeLeve, Wang et al. 2004, Xie, Choi et al. 2013, Desroches-Castan, Tillet et al. 2019). So, in order to study these fenestrae, one has to work on freshly isolated LSEC that need first to adhere in order to perform SEM. Most of the publications, thus use an overnight culture time (around 18 hours) but some have used very early time points such as 3 hours after cell culture (Xie, Choi et al. 2013). We first analyzed LSEC derived from C57BL/6 mice after 18h of cultures but we were surprised to see a very low number of fenestrae as compared to LSEC derived from 129/Ola mice thus we decided to look at earlier time points so that we could have more fenestrae for quantification. Still, as presented in the manuscript we found a much lower number of fenestrae in C57BL/6 LSEC compared to 129/Ola LSEC even at this earlier timepoint (Fig3A and Supp Fig1). We have added a reference to make this point clearer.

The error bars in Fig. 3 are quite large and significance cannot be determined. Consider repeating experiments.

The error bars are indeed large but the means are quite similar between WT and Bmp9-KO mice so we do not think that some of these genes will become differently regulated if we increase the number of experiments. With the limitation of time given for our review (10 days) it is not possible to repeat the experiments that would need mouse experiments. Still, we analyzed these data using also the data from figure 4 from WT C57BL/6 mice (WT=9 instead of 5 in Fig3B, supplementary points from figure 4 are shown in red). However, it did not lead to significant differences between WT and Bmp9-KO mice (see figure for the reviewer).

DeLeve, L. D., X. Wang, L. Hu, M. K. McCuskey and R. S. McCuskey (2004). "Rat liver sinusoidal endothelial cell phenotype is maintained by paracrine and autocrine regulation." Am J Physiol Gastrointest Liver Physiol287(4): G757-763.

Desroches-Castan, A., E. Tillet, N. Ricard, M. Ouarne, C. Mallet, L. Belmudes, Y. Coute, O. Boillot, J. Y. Scoazec, S. Bailly and J. J. Feige (2019). "Bone Morphogenetic Protein 9 Is a Paracrine Factor Controlling Liver Sinusoidal Endothelial Cell Fenestration and Protecting Against Hepatic Fibrosis." Hepatology.

Xie, G., S. S. Choi, W. K. Syn, G. A. Michelotti, M. Swiderska, G. Karaca, I. S. Chan, Y. Chen and A. M. Diehl (2013). "Hedgehog signalling regulates liver sinusoidal endothelial cell capillarisation." Gut62(2): 299-309.

Reviewer 3 Report

This manuscript is from an established expert group working on BMP9, primarily using mouse studies. Although the abstract states that the aim of this work is to address the role of BMP9 in different genetic backgrounds (C57BL/6, BALB/c and 129/Ola) I am uncertain that this goal has been met. The work describes different penetration of a liver fibrosis phenotype in BMP9 knockout mice depending on genetic background, with liver fibrosis present in 129/ola but not C57BL/6 or Balb/C mice. The remaining experiments show further differences in the liver or liver endothelial cells from 129ola and BL6 mice but conclusions drawn relating BMP9 to different liver fibrosis outcomes remain speculative.

1.       Work has been ethically approved, but is death as an endpoint an ethically approved measure for the large number of mice used? Or were more humane end points used for these animals?

2.       What were the causes of death? Is there evidence for loss of liver or kidney function? Were any organ function tests performed?

3.       The analysis of Collagen IV staining in the liver sinusoids is confusing, data is only shown for BL6 (Fig3c &D) but the conclusion drawn (line 371-2) is that this protein is present at reduced levels in 129/ola. Where is the data?

4.       Fig3 and 4 qPCR data need to be combined and rationalised as the data currently provides conflicting conclusions. For example Stab2 levels are 3 times greater than Stab 1 levels in BL6 LSECs in Fig3, but this ratio is completely reversed in Fig4. This reduces confidence in these data and its interpretation.

5.       How was the sinusoidal surface measured in Fig 2C &D? Can this be illustrated using higher power images? Also how can the authors discriminate between what is tissue morphology and what is dehydration artefact?  

6.       Supp Fig 1B please show how fenestrae were defined and counted. The authors point out that LSEC fenestrae alter during culture, so were any findings confirmed in vivo? As the authors propose the reduced fenestrae is due to lower BMP9 levels in BL6 did they attempt to test this idea eg to test for increased fenestrae with BMP9 treatment?

7.       Is there any evidence (other than association) that reduced basal BMP9 levels lead to any of the reported changes in 129ola compared with BL6 mice? 

8.       More specifically, the relevance of basal BMP9 differences to the altered liver pathology is unclear. How do higher basal BMP9 levels in 129/ola compared with BL6 mice make 129/ola more sensitive to liver fibrosis when BMP9 is lost? This is hard to rationalise with published data showing that BMP9 is pro-fibrogenic. Can the authors provide any explanation?

9.       Statistical analyses use non-parametric analyses (Mann Whitney) but data are plotted as mean +/- error bars (SEM or SD?) which is traditionally used to represent a normal distribution of data. This is confusing.

Minor Points

The split summary of findings between line 51 and 70 of the introduction is confusing.

Mention of PTPN14 modifier of HHT should also be in the introduction.

Round 2

Reviewer 3 Report

The presentation of the work is improved. However, data from Ref 15 is required for the reader to fully understand and interpret the data presented in this manuscript. It is of course appropriate to reference previous findings but it is not possible to accurately compare the current data with that in ref 15 (eg collagen IV staining) because experiments were not performed in parallel. Also, LSEC differences were only seen in vitro and not in vivo. These limitations should be acknowledged in the discussion. 

Author Response

We have performed a new Collagen IV immunostaining in liver sections of WT and Bmp9-KO mice from C57BL/6 and 129/Ola mice in parallel on the same day under the same conditions. The results, which confirm our previous statements are shown in Fig3C and Supp Fig1C, respectively.

We now have specified in the discussion section that LSEC analyses were performed on freshly isolated cells thus not in vivo butex vivo
